# Low Back Pain Prevalence among Distance Learning Students

**DOI:** 10.3390/ijerph20010342

**Published:** 2022-12-26

**Authors:** Mohannad Hawamdeh, Thamer A. Altaim, Amjad Shallan, Riziq Allah Gaowgzeh, Sakher M. Obaidat, Saad Alfawaz, Saad M. Al-Nassan, Ziyad Neamatallah, Owis Eilayyan, Umar M. Alabasi, Majed Albadi

**Affiliations:** 1Department of Physical and Occupational Therapy, Faculty of Applied Medical Sciences, The Hashemite University, P.O. Box 330127, Zarqa 13133, Jordan; 2Physical Therapy Department, Faculty of Allied Medical Sciences, Aqaba University of Technology, Aqaba 77110, Jordan; 3Department of Physical Therapy, Faculty of Medical Rehabilitation Sciences, King Abdulaziz University, Jeddah 21589, Saudi Arabia; 4Department of Physical Therapy, Faculty of Allied Health Sciences, Al-Ahliyya Amman University, Amman 19328, Jordan

**Keywords:** low back pain, students, online learning, COVID-19, posture

## Abstract

Background: Low back pain as a symptom affects many individuals around the globe regardless of their economic status or sociodemographic characteristics. During the 2019 COVID-19 pandemic, students found themselves obligated to sit down for long periods of time. The aim of this current study is to investigate the impact of these prolonged periods of sitting down in front of computers on developing a new episode of low back pain. Methods and Materials: This research adopted an observational cross-section study design. Students who are currently enrolled or had experienced distance learning classes in the last 6 months were eligible to participate. An online-based questionnaire was developed by the investigators through reviewing the literature with relevant objectives. McNemar’s test was used to compare certain variables between two periods before and during online distance learning. We used paired *t*-tests to compare pain intensity before, during, and after online learning, while a chi-square test was used to investigate correlations between factors influencing low back pain. Results: A total of 84 students participated in the study—46 (54.8%) females and 38 (45.2%) males. Before online distance learning, only 42.9% of participants reported low back pain, while only 20% had a back injury. The mean pain scores before, during, and after online distance learning were (2.85 ± 2.16, 4.79 ± 2.6, and 4.76 ± 2.7), respectively. The pain scores before online learning were significantly lower than pain scores during and after online distance learning (*p* < 0.05), respectively. Conclusion: The study findings suggested that low back pain prevalence increased among students during the COVID-19 pandemic. Future research should study participants’ behavior during the online learning and assess the long-run impact of distance learning among high-school and undergraduate students.

## 1. Introduction

Low back pain (LBP) is considered one of the most common musculoskeletal disorders that affect people of all ages around the world [1]. In addition, LBP is becoming a burden in many developing and low-income countries due to high healthcare costs for its treatment and management [2]. Prolonged sitting and computer use in the workplace, in addition to improper postures, may contribute to low back pain [2].

During the COVID-19 pandemic, many governments worldwide made a series of decisions to control the pandemic’s spread, such as quarantine, social distancing, and switching from traditional education to different forms of E-learning [3,4]. E-learning is referred to learning by using specific technologies and it is considered a type of distance learning, as opposed to traditional classroom learning [5]. E-learning has some advantages such as students having more flexibility in terms of choosing the place and time for learning, as well as cost effectiveness for both students and education systems [6]; however, E-learning has many disadvantages that should be addressed. For example, E-learning may decrease communication between students and teachers [7]. Additionally, E-learning methods usually require students to stay sitting for a long time on their laptops and smart devices, which may lead to different health problems [8].

Many studies have found a strong correlation between using advanced technology, such as smartphones, and spine disorders [9,10,11]. A systematic review was conducted to evaluate the impact of using smart devices on musculoskeletal complaints, and it showed there was a strong relationship between the duration smartphone use and spine disorders [12]. Moreover, most students are seated on non-ergonomic tables and chairs during E-learning, which may lead to bad postures, and thus could lead to some problems in the lumbar spine region such as stiffness and abnormal muscle activation [13,14,15]. Several studies found that students who spent more than 3 h per day on their laptops suffered from neck and back pain; however, this musculoskeletal pain was more evident in inactive students compared to active ones [16,17]. Moreover, several studies found that there are many factors that could contribute to musculoskeletal pain in university students engaging in E-learning methods, including psychosocial and environmental factors [18]. These factors could contribute to musculoskeletal pain either directly or indirectly [6].

Usually, university students have little knowledge regarding effective strategies for pain management, which may lead to a negative impact on their musculoskeletal pain [19]. In addition, university students with musculoskeletal pain such as low back pain may experience some interference with their academic performance [19]. Therefore, attention should be paid to address this issue, especially in Middle East countries, where the age-standardized point prevalence of LBP was highest at 14.8% [20]. However, to our knowledge, few studies have been conducted to measure the impact of E-learning on LBP among university students in developing and low-income countries. Therefore, the aim of this study was to investigate the prevalence of low back pain among university students attending virtual classes during the COVID-19 pandemic.

## 2. Materials and Methods

### 2.1. Design and Population of the Study

This was a cross-sectional observational study. All students at high school level or university level who are currently enrolled or had experienced distance learning classes in the last 6 months were eligible to participate. Participation was exclusive for students at Jordanian universities and high schools.

### 2.2. The Questionnaire

An online-based questionnaire was developed by the investigators through reviewing the literature with relevant objectives; the questionnaire used and revised by experts was inspired by Abbas et al. (2021) [21]. A pilot test was performed on a small group of university students to test the clarity, ease, and time needed to complete the questionnaire. The questionnaire required about 15 min to complete. A panel of experts in the fields of physical therapy and rehabilitation reviewed the questionnaire and provided their feedback regarding the face and content validity of the included items and questions. The final version of the questionnaire was developed electronically through Google Forms and included 26-closed-ended items divided into 3 main parts. The first part (5 items) included basic demographic questions, the second part (4 items) asked about lifestyle and daily habits, while the third part (17 items) included general and low back pain-related questions. An electronic link to the questionnaire was shared with the participants through social media networks. A cover letter of the questionnaire was enclosed within the link of the questionnaire to ensure eligibility of participation and a pledge of confidentiality of participants’ information. The questionnaire was delivered in the Arabic language (Appendix A), and the study protocol was approved by the institutional review board at The Hashemite University.

### 2.3. Statistical Analysis

Statistical analysis was performed using the Statistical Package for the Social Sciences (SPSS) Software version 25 for Windows (Chicago, IL, USA). Mean and standard deviation for quantitative variables and counts (%) for qualitative variables were used to summarize data. The normality of continuous variables was examined using Shapiro–Wilk test. The level of significance was set at *p* ≤ 0.05.

## 3. Results

### 3.1. Demographic Data

A total of 84 students participated in the study—46 (54.8%) females and 38 (45.2%) males. Most of the students were studying bachelor’s degrees (86.9%); 79.8% were in the age range of 18 to 22. Participants’ BMIs were categorized into four categories: underweight (BMI < 18.5), healthy weight (BMI 18.5–24.9), overweight (BMI 25–29.9), and obese (BMI > 30). We found that 51.1% of students were within the healthy weight range (see Table 1).

### 3.2. Lifestyle before and during Online Distance Learning

Before online distance learning, 26.2% of students did not exercise at all. During online distance learning, 35.7% did not exercise at all. However, this difference was not statistically significant (*p* = 0.115) (see Table 2). Our results found that students used to spend an average of 7.7 h daily using electronic devices during online distance learning, where most of them (51.2%) reported that they spent more than 6 h a day using electronic devices (see Table 2).

The high usage of electronic devices during online distance learning could be attributed not only to using devices for attending lectures and studying but also for watching TV or spending time via social media.

### 3.3. Low Back Pain Incidence and Intensity before and during Online Distance Learning

Before online distance learning, only 42.9% (see Table 3) of participants reported low back pain, while only 20% had a back injury. The mean pain scores before, during, and after online distance learning were (2.85 ± 2.16, 4.79 ± 2.6, and 4.76 ± 2.7), respectively. The pain scores before online learning were significantly lower than pain scores during and after online distance learning (*p* < 0.05), respectively (see Table 4).

The most-reported pain was stiffness (40.5%), while the lowest-reported was burning pain (4.8%). Approximately 65.5% of participants reported that low back pain was not hindering their movement; at the same time, 76.2% of participants reported that they did not take any pain relief medications (see Table 3). In total, 65.5% of participants reported that they have been infected with COVID-19, where a 40% reported the continuation of low back pain after getting COVID-19 infection; most of the participants 40% reported that their pain level remained the same (see Table 3).

### 3.4. Sleep and Psychological Stress during Online Distance Learning

We categorized sleep hours into two groups: less than 8 h and more than 8 h. We found that most of participants (61.9%) spent less than 8 h of sleep during the period of online learning. Furthermore, 76.2% of participants reported an experience of psychological stress during online distance learning (see Table 5). Table 6 present the association between the occurrence of LBP and some of the demographics characteristics of the participants.

### 3.5. Posture during Online Distance Learning

We found that most participants did not use the proper posture to attend lectures. Only 36.9% used desks to attend online lectures, while the remaining participants reported lying on a bed 32.1%, sitting on a couch 17.9%, or using an Arabic mattress 13.1% (see Table 7).

### 3.6. Association between Low Back Pain and Demographics and Other Parameters

Gender, educational level, and BMI were not significantly associated with low back pain. There was no significant association between low back pain and the time using electronic devices, exercising during online learning, number of sleep hours, and posture during online lectures. However, exercising before online learning (OR = 0.314, 95% CI [0.114–0.865], *p* = 0.025) was independently associated with a low prevalence of LBP at the time of the study (see Table 7).

## 4. Discussion

### 4.1. Online Learning and Incidence of LBP

This study aimed to estimate the prevalence of low back pain among university students attending virtual classes during the COVID-19 pandemic. Some demographic and lifestyle-related factors were assessed in this study using a content validated questionnaire. The results showed that the prevalence and the intensity of LBP increased during distance learning throughout COVID-19 pandemic. This is supported by the literature, where the percentage of people who suffered from LBP increased significantly during COVID-19 and pain intensity also increased among people who underwent distance learning [22]. In addition, most of the participants in this study reported stiffness. This might be related to the fact that most of the participants did not use a proper posture and did not use the ergonomic chairs and desk during online learning. This may increase tension in the back muscles and cause stiffness.

Although one-third of the participants were infected with COVID-19, this study showed that there was no correlation between having COVID-19 and the incidence of LBP. This is supported in the literature. For instance, the WHO shows that aches and pains are less-common symptoms among people with COVID-19 [23]. Additionally, Murat et al. showed that only 28% of the participants who suffered from COVID-19 reported back pain [24]. Furthermore, the literature supports that during online learning, as a result of COVID-19, students spent more time using the laptops, computers and smart devices, which increased the incidence and intensity of musculoskeletal pain, including LBP [25,26,27]. The results of this study showed that the average sleeping hours and average hours of using electronic devices did not significantly affect LBP, which is supported by Basri et al.’s study, where they found no correlation between LBP with sleeping and smartphone usage per day [28]. Lastly, there was no significant difference in the prevalence of LBP in relation to gender, which is consistent with other studies [22,29].

### 4.2. Posture/Physical Activity and LBP

Prolonged sitting during online learning may worsen LBP [30], as it may decrease low back muscle activation; this leads to a reconditioning of the muscles and increasing the load at intervertebral and ligaments [31]. Furthermore, a prolonged static posture may increase physical stress, which in turn can injure soft tissues and cause musculoskeletal pain and LBP [27,32,33]. The participants in this study used different ways of sitting, such as a comfortable position, to reduce tension on the low back area, and this is why posture during online learning was not a significant factor of LBP. This is supported in the literature, where sitting hours and awkward posture were not correlated with LBP [28]. However, one study showed that people who adhered to ergonomic recommendations had significantly less pain [22], and other studies have reported that prolonged sitting on a chair may cause joint pain [34,35,36]. A lack of physical activity has an important role in increasing the intensity of LBP [37]. During online learning, students need to spend more time in a sitting position, and this may reduce physical activity [26]. However, the majority of participants in the current study kept performing exercise during online learning. This may explain why physical activity did not significantly affect LBP. A study conducted in 2022 among undergraduate students showed that self-perceived exercise was not significantly correlated with LBP [38], and another study showed that not performing sports activities during the COVID-19 pandemic was not associated with LBP [28].

### 4.3. Psychological Issues

The majority of participants reported psychological issues. Mattioli et al. reported that quarantine increased psychological distress among people, which pushed people to adopt unhealthy lifestyles and worsen LBP [39]. Online learning was shown to result in academic stress, as it increases the workload and is not suitable for courses that require hands-on skills [40]. This may affect the mental status of students. Many studies show that psychological distress may increase the intensity of pain among people suffering from LBP [41,42,43,44]. Sagat et al. (2020) showed that stress increased the intensity of LBP among Saudi people during COVID-19 quarantine [22]. However, the results of this study showed that psychological stress was not a significant factor in LBP, and might instead be related to the fact that most of the participants kept doing exercises during online learning.

### 4.4. Future Consideration

This study assessed many factors that could affect LBP. However, none of the studied variables significantly contributed to the incidence or the severity of LBP among Jordanian students. The next step is to study participants’ behavior during online learning and assess the long-run impact of distance learning among high-school and undergraduate students in order to target significant variables and improve health status among students.

### 4.5. Limitations

There were some limitations in this study that need to be noted. The number of the participants in this study was not high; only 84 students participated. Therefore, the results of this study cannot be generalized to all students in Jordan. Additionally, there were some variables that were not assessed in the study, such as pre-exercise and sedentary levels and psychological distress before COVID-19.

## 5. Conclusions

We concluded that the majority of the students spent the same number of hours using electronic devices—an average of 7.7 h daily—during online distance learning, and most of them (51.2%) reported that they spent more than 6 h a day using electronic devices. However, we found that the pain scores before online learning were lower than during and after online distance learning. The results showed that the prevalence and intensity of LBP increased during distance learning throughout the COVID-19 pandemic. The next step is to study participants’ behavior during online learning and assess the long-term impact of distance learning among high-school and undergraduate students in order to target the significant variables and improve health status among students.

## Figures and Tables

**Table 1 ijerph-20-00342-t001:** Demographic data.

	Total *n* (%)
Gender	84
Female	46 (54.8%)
Male	38 (45.2%)
Educational level	
High School	5 (6.00%)
Diploma	4 (4.8%)
Undergraduate BSc	73 (86.9%)
Graduate	2 (2.4%)
BMI level	
Underweight	5 (6.00%)
Healthy weight	43 (51.2%)
Overweight	28 (33.3%)
Obese	8 (9.5%)
Age groups	
18–22	67 (79.8%)
23–28	12 (14.3%)
>28	5 (6.00%)

Abbreviations: BMI, Body mass index.

**Table 2 ijerph-20-00342-t002:** Lifestyle before and during online distance learning.

	Before Online Learning *n* (%)	During Online Learning *n* (%)	*p*-Value
Exercise			0.115
Yes	62 (73.8%)	54 (64.3%)	
No	22 (26.2%)	30 (35.7%)	
Electronic devices hours			
<6		41 (48.8%)	
<6		43 (51.2%)	

Abbreviations: *p*, Probability value (according to McNemar test).

**Table 3 ijerph-20-00342-t003:** Low back pain prevalence, duration, and nature, and associated disabilities.

	*n* (%)
Low back pain	
Yes	48 (57.1%)
No	36 (42.9%)
Duration of pain	
Discontinuous	67 (79.8%)
No pain	12 (14.3%)
Permanent	5 (6.00%)
Nature of pain	
Burning pain	4 (4.8%)
Like a shock	4 (4.8%)
Like an appeal	11 (13.1%)
no pain	11 (13.1%)
numbness	6 (7.1%)
prick	14 (16.7%)
stiffen	34 (40.5%)
Pain hindering movement	
Yes	29 (34.5%)
No	55 (65.5%)
Pain relief medications	
Yes	20 (23.8%)
No	64 (76.2%)
Physiotherapy for LBP	
Yes	11 (13.1%)
No	73 (86.9%)
COVID-19 infection	
Yes	29 (34.5%)
No	55 (65.5%)

Abbreviations: BMI, Body mass index; LBP, LowbBack pain.

**Table 4 ijerph-20-00342-t004:** Low back pain severity before, during, and after online distance learning.

	Before Online Learning Mean ± SD	During Online Learning Mean ± SD	After Online Learning Mean ± SD	*p*-Value
LBP severity	2.85 ± 2.16	4.79 ± 2.6	4.76 ± 2.69	P1 = 0.000 *
				P2 = 0.000 *

Abbreviations: LBP, Low Back Pain; *p*, Probability value (according to paired *t*-test); P1, refers to the difference in LBP severity before to during online learning; P2, refers to the difference in LBP severity before and after online learning; *, statistically significant difference.

**Table 5 ijerph-20-00342-t005:** Sleep hours and psychological stress after online distance learning.

	*n* (%)
Sleep hours’ number	
<8	52 (61.9%)
>8	32 (38.1%)
Psychological stress	
Yes	64 (76.2%)
No	20 (23.8%)

**Table 6 ijerph-20-00342-t006:** Association between low back pain and demographics.

	Low Back Pain *n* (%)	No Low Back Pain *n* (%)	OR (95% CI)	*p*-Value
Gender				
Female	19 (52.8%)	27 (56.2%)	1 ^	
Male	17 (47.2%)	21 (43.8%)	1.15 [0.43–2.74]	0.752
Educational level				
High School	1 (2.8%)	4 (8.33%)	1 ^	
Diploma	1 (2.8%)	3 (6.25%)	1.33 [0.057–31.121]	0.858
Undergraduate BSc	33 (91.6%)	40 (83.33%)	3.30 [0.352–30.9]	0.296
Graduate	1 (2.8%)	1(2.09%)	4.00 [0.117–136.957]	0.442
BMI level				
Underweight	3 (8.3%)	2 (4.2%)	1 ^	
Healthy weight	14 (38.8%)	29 (60.4%)	0.322 [0.048–2.151]	0.242
Overweight	14 (38.8%)	14 (29.2%)	0.667 [0.096–4.62]	0.682
Obese	5 (13.8%)	3 (6.3%)	1.11 [0.112–10.98]	0.928
Age groups				
18–22	28 (8.3%)	39 (8.3%)	1 ^	
23–28	4 (38.8%)	8 (38.8%)	0.696 [0.191–2.542]	0.584
>28	4 (38.8%)	1 (38.8%)	5.57 [0.096–52.567]	0.134

Abbreviations: OR Odds Ratio; CI, Confidence Interval; *p*, Probability value (according to logistic regression); ^, reference category.

**Table 7 ijerph-20-00342-t007:** Association between low back pain and physical parameters.

	Low Back Pain *n*%	No Low Back Pain *n*%	OR (95% CI)	*p*-Value
Exercise before online learning				
Yes	22 (61.1%)	40 (83.3%)	1 ^	
No	14 (38.9%)	8 (16.7%)	0.314 [0.114–0.865]	0.025 *
Exercise during online learning				
Yes	22 (61.1%)	32 (66.6%)	1 ^	
No	14 (38.9%)	16 (33.4%)	0.786 [0.320–1.932]	0.599
Hours of using electronic devices				
<6	16 (44.4%)	25 (52.08%)	1 ^	
>6	20 (55.6%)	23 (47.92%)	1.359 [0.571–3.23]	0.488
Posture during online lectures				
Lying on Bed	13 (36.1%)	14 (29.2%)	1 ^	
Sitting at the desk	13 (36.1%)	18 (37.5%)	0.778 [0.275–2.19]	0.635
Sofa-couch sitting	5 (13.8%)	10 (20.1%)	0.538 [0.145–2.00]	0.355
Arabic mattress sitting	5 (13.8%)	6 (12.5%)	0.897 [0.220–3.66]	0.880
COVID-19 infection				
Yes	24 (66.7%)	31 (64.6%)	1 ^	
No	12 (33.3%)	17 (35.4%)	1.097 [0.441–2.728]	0.842
Psychological stress				
Yes	29 (80.5%)	35 (72.9%)	1 ^	
No	7 (19.5%)	13 (27.1%)	1.53 [0.543–4.364]	0.418
Hours of sleep				
<8	20 (55.6%)	32 (66.7%)	1 ^	
>8	16 (44.4%)	16 (33.3%)	1.6 [0.657–3.89]	0.301

Abbreviations: OR Odds Ratio; CI, Confidence Interval; *p*, Probability value (according to logistic regression); *, statistically significant difference; ^, reference category.

## Data Availability

Not applicable.

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
