# Peer review of "Low Back Pain Prevalence among Distance Learning Students"

_ijerph, 2022, doi:10.3390/ijerph20010342_

Round 1
Reviewer 1 Report
The authors present an article investigating the effects of distance learning on low back pain prevalence in students. This area has received a little attention in the literature, therefore, warrants further examination. Overall, the manuscript follows the logical sequence of a research purpose. However, I have comments that need to be addressed to the authors and listed below.
ABSTRACT.
Results should at least present p-values. Conclusion should be more precise and introduce future research related to the results.
INTRODUCTION
Introduction should be improved and more developed. Some key papers related to the aim of this study are missing (listed below). In addition, some of them should be used in the discussion.
Ayyildiz E, Taskin Gumus A. A novel distance learning ergonomics checklist and risk evaluation methodology: A case of Covid-19 pandemic. Hum Factors Ergon Manuf. 2021; 31(4):397-411. doi: 10.1002/hfm.20908.
Kim HJ, Boo S, Meeker TJ. Pain prevalence, management and interference among university students in South Korea: An exploratory cross-Sectional study. J Pain Res. 2021, 14:2423-2431. doi: 10.2147/JPR.S324758.
Memari A, Shariat A, Anastasio AT. Rising incidence of musculoskeletal discomfort in the wake of the COVID-19 crisis. Work. 2020, 66(4):751-753. doi: 10.3233/WOR-203221.
Roggio F, Trovato B, Ravalli S, Di Rosa M, Maugeri G, Bianco A, Palma A, Musumeci G. One Year of COVID-19 Pandemic in Italy: Effect of sedentary behavior on physical activity levels and musculoskeletal pain among university students. Int J Environ Res Public Health. 2021, 18(16):8680. doi: 10.3390/ijerph18168680.
Salameh MA, Boyajian SD, Odeh HN, Amaireh EA, Funjan KI, Al-Shatanawi TN. Increased incidence of musculoskeletal pain in medical students during distance learning necessitated by the COVID-19 pandemic. Clin Anat. 2022, 35(4):529-536. doi: 10.1002/ca.23851.
Silva AG, Sa-Couto P, Queirós A, Neto M, Rocha NP. Pain, pain intensity and pain disability in high school students are differently associated with physical activity, screening hours and sleep. BMC Musculoskelet Disord. 2017, 18(1):194. doi: 10.1186/s12891-017-1557-6.
Silva GR, Pitangui AC, Xavier MK, Correia-Júnior MA, De Araújo RC. Prevalence of musculoskeletal pain in adolescents and association with computer and videogame use. J Pediatr (Rio J). 2016, 92(2):188-96. doi: 10.1016/j.jped.2015.06.006.
Vitta A, Bento TPF, Cornelio GP, Perrucini PDO, Felippe LA, Conti MHS. Incidence and factors associated with low back pain in adolescents: A prospective study. Braz J Phys Ther. 2021, 25(6):864-873. doi: 10.1016/j.bjpt.2021.10.002.
MATERIALS AND METHODS
This section should be improved. Authors should present several subsections: 1) population of the study related to table 1, 2) questionnaire and, 3) statistical analyses.
Concerning the questionnaire used, the questionnaire should be added in a supplementary file. The authors mentioned that the questionnaire has been tested in a pilot phase according to expert ‘feedback. Which experts? And should we understand that there is no validation of the questionnaire done? If yes, this is a serious issue for the reliability of the results.
RESULTS
This section should be improved. Authors should refer their results to tables. F values are missing and should be added. Please, use Body mass index first then the abbreviation BMI.
Subsections should be better identified :
3.1 demographic data (some information should be remove in Materials and Methods section)
3.2 Lifestyle before and during online distance learning
3,3 Low back pain incidence and intensity before and during online distance learning (in this subsection, please rephrase the sentences “But when we compared mean pain scores before, during, and 104 after online distance learning were (2.85±2.16, 4.79±2.6, and 4.76±2.7) respectively. The 105 pain scores before online learning were significantly lower than pain scores during and 106 after online distance learning (p<.05) respectively” for a better understanding).
3.4 Sleep and psychological stress during online distance learning
3.5 Posture during online distance learning
TABLES
All tables should be improved and some merged.
Please change "n%" by "n(%)" in first line for each tables when appropriated.
Table 1 and 6 should be merged (total sample in a column, no low back pain in another column and low back pain in another column).
Table 4. First line, change in column 4th “during online learning” by “after online learning”. P1 and P2 are not referred. Seems that severity of the low back pain is an ordinal score expressed by the mean ±SD, why “n%” used here?
Table 6. P values are insufficient, 95% CI should be added.
DISCUSSION
This section should be improved. Authors should discuss the limits of their study (for example, development and validation of the questionnaire) and better linked their results with those observed in the literature.
Their current conclusion is a point to insert in the discussion. The discussion should focus on the main results of their study and present future needed research or recommendation.
REFERENCES
Please verify all references (check the norm of the journal for their presentation). Moreover, many references are incorrect or incomplet For example, references 1 to 13, journal name is missing. Or reference 8 should be: "Kim HJ, Kim JS. The relationship between smartphone use and subjective musculoskeletal symptoms and university students. J Phys Ther Sci. 2015, 27(3):575-9".
Author Response
Manuscript title: Low Back Pain Prevalence Among Distance Learning Students
Response to the reviewers’ comments:
Thank you for your time and comments on this manuscript. We found these comments helpful and will improve the clarity and readability of this research article. Please note that within the following paragraphs we presented a reply to each of the reviewers’ comments. We provided a point-by-point response to the reviewer’s comments within the following subsections.
Reviewer 1:
Reviewer 1: The authors present an article investigating the effects of distance learning on low back pain prevalence in students. This area has received a little attention in the literature, therefore, warrants further examination. Overall, the manuscript follows the logical sequence of a research purpose. However, I have comments that need to be addressed to the authors and listed below.
Response: We would like to thank Reviewer 1 for his/her time and comments which helped in improving the overall quality of this manuscript.
Abstract
Reviewer 1: Results should at least present p-values. Conclusion should be more precise and introduce future research related to the results.
Response: P-values were added in the result subsection. The paragraph was amended to read the following “Before online distance learning only 42.9% of participants reported low back pain, while only 20% had back injury. The mean pain scores before, during, and after online distance learning was (2.85±2.16, 4.79±2.6, and 4.76±2.7) respectively. The pain scores before online learning were significantly lower than pain scores during and after online distance learning (p<.05) respectively.” The conclusion subsection was also amended. We rewrote it to read the following “The study findings suggested that low back pain prevalence increased among students during the COVID 19 pandemic. Future research should study participants’ behavior during online learning and assess the long-run impact of distance learning among high-school and undergraduate students.”
Introduction
Reviewer 1: Introduction should be improved and more developed. Some key papers related to the aim of this study are missing (listed below). In addition, some of them should be used in the discussion.
Response: We thank the reviewer very much for his/her suggestions and comments. We amended the introduction and added more references as requested. Please see these amendments in the introduction and discussion sections.
Materials and Methods
Reviewer 1: This section should be improved. Authors should present several subsections: 1) population of the study related to table 1, 2) questionnaire and, 3) statistical analyses.
Response: We thank the reviewer for his/her comment. As requested, we amended this section and add the following subsections:
- Population subsection in page 2
- Questionnaire subsection in page 2
- Statistical analysis in page
Reviewer 1: Concerning the questionnaire used, the questionnaire should be added in a supplementary file. The authors mentioned that the questionnaire has been tested in a pilot phase according to expert ‘feedback. Which experts? And should we understand that there is no validation of the questionnaire done? If yes, this is a serious issue for the reliability of the results.
Response: We thank reviewer 1 for his/her helpful comments. As requested, we added a supplementary file that contains the questionnaire used in this study. The questions used in this study were used before in another study by Abbas et al. (Abbas J, Hamoud K, Jubran R, Daher A. Has the COVID-19 outbreak altered the prevalence of low back pain among physiotherapy students? Journal of American College Health. 2021:1-6.). With regard to the expert committee who were involved in this study, they were assistant professors in the physical therapy department at Hashemite University in Jordan. We added the ‘Questionnaire’ subsection and added a paragraph that addresses the reviewer's comments.
Results
Reviewer 1: This section should be improved. Authors should refer their results to tables. F values are missing and should be added. Please, use Body mass index first then the abbreviation BMI.
Response: We rewrote this section and referred to the tables within the result section. F values were added. We spelled out all abbreviations before using them again in the following paragraphs.
Reviewer 1: Subsections should be better identified.
Response: We followed the reviewer's comment and identified subsections according to the reviewer's instructions.
Tables
Reviewer 1: All tables should be improved and some merged.
Response: We amended the tables according to the reviewer's comments.
Reviewer 1: Please change "n%" by "n(%)" in first line for each tables when appropriated.
Response: We changed "n%" to "n(%)" in the first line for each table.
Reviewer 1: Table 1 and 6 should be merged (total sample in a column, no low back pain in another column and low back pain in another column).
Response: We merged Tables 1 and 6 as requested.
Reviewer 1: Table 4. First line, change in column 4th “during online learning” by “after online learning”. P1 and P2 are not referred. Seems that severity of the low back pain is an ordinal score expressed by the mean ±SD, why “n%” used here?
Response: We changed Table 4 as per reviewer instructions. We deleted mean±SD.
Reviewer 1: Table 6. P values are insufficient, 95% CI should be added.
Response: We added Confidence Interval to Table 6.
Discussion
Reviewer 1: This section should be improved. Authors should discuss the limits of their study (for example, development and validation of the questionnaire) and better linked their results with those observed in the literature.
Response: Thank you for your comment, a limitation section was added to the end of the discussion section. Also, a paragraph was added to the discussion to link between our results and the ones available in the literature.
Reviewer 1: Their current conclusion is a point to insert in the discussion. The discussion should focus on the main results of their study and present future needed research or recommendation.
Response: Future direction and recommendations were added to the discussion section
References
Reviewer 1: Please verify all references (check the norm of the journal for their presentation). Moreover, many references are incorrect or incomplete. For example, references 1 to 13, journal name is missing. Or reference 8 should be: "Kim HJ, Kim JS. The relationship between smartphone use and subjective musculoskeletal symptoms and university students. J Phys Ther Sci. 2015, 27(3):575-9".
Response: The reference list was checked and all the amendments suggested by the reviewer were addressed.

Reviewer 2 Report
Thank you to the authors for their submission. The study is basic however quantifies the impact on LBP of distance learning. The article has merit, however requires extensive English editing and additions to results, discussion, and conclusion. Please see edits below.
1. I have used strikethrough and comments to suggest edits and replacement words in the attached document. Please fix through. I have done this only for the introduction, but the discussion needs similar work with a native English speaking editor.
2. Please spell out all acronyms at first usage - I have highlighted the acronyms that need to be spelled out in green highlight in the attached document.
3. Please put the city and country for SPSS and version number.
4. The results section needs sub-headings to organise the different descriptions of data - i.e. physical activity, sex, BMI, etc.
5. You need to provide the test statistic for each of the test conducted, not just the p value. Please amend the tables accordingly.
6. Below each table you need a footnote explaining all abbreviations, the test statistic, and interpretations of each measure (i.e. score range and if higher/lower score is better/worse).
7. You need to provide the questionnaire used as a supplement for verification. There needs to be much more analysis of the use of an unvalidated questionnaire.
8. Your discussion would benefit from using sub-headings to organise your thoughts. Please place in subheadings defining the topic of your argument and improve the linkage between arguments.
9. You need to include a limitations section - for which this study has some major limitations. These include, but are not limited to, lack of control group, missing pre-exercise and sedentary levels, unvalidated questionnaire, etc.
10. The conclusion need to mentioned more about future directions and areas for exploration.
Thank you.

Author Response
Manuscript title: Low Back Pain Prevalence Among Distance Learning Students
Response to the reviewers’ comments:
Thank you for your time and comments on this manuscript. We found these comments helpful and will improve the clarity and readability of this research article. Please note that within the following paragraphs we presented a reply to each of the reviewers’ comments. We provided a point-by-point response to the reviewer’s comments within the following subsections.
Reviewer 2:
Reviewer 2: Thank you to the authors for their submission. The study is basic however quantifies the impact on LBP of distance learning. The article has merit, however requires extensive English editing and additions to results, discussion, and conclusion. Please see edits below.
Response: We would like to thank the reviewer for his time and comments which increased the quality of this manuscript and enhanced its readability. We carried out all suggested edits and additions to the results, discussion, and conclusion sections.
Reviewer 2: I have used strikethrough and comments to suggest edits and replacement words in the attached document. Please fix through. I have done this only for the introduction, but the discussion needs similar work with a native English speaking editor.
Response: Thank you very much for your time and comments. We reviewed these suggestions and carried out amendments within the discussion section.
Reviewer 2: Please spell out all acronyms at first usage - I have highlighted the acronyms that need to be spelled out in green highlight in the attached document.
Response: We spelled out all abbreviations at their first use and performed all amendments highlighted by Reviewer 2.
Reviewer 2: Please put the city and country for SPSS and version number.
Response: We added the following statement “IBM SPSS Statistics for Windows, Version 26.0. [Armonk, NY, USA], was used for data analysis”.
Reviewer 2: The results section needs sub-headings to organise the different descriptions of data - i.e. physical activity, sex, BMI, etc.
Response: We amended the result section to include the suggested subsections.
Reviewer 2: You need to provide the test statistic for each of the test conducted, not just the p value. Please amend the tables accordingly.
Response: The test statistics were provided in each of the tables within the result section per Reviewer 2 comment.
Reviewer 2: Below each table you need a footnote explaining all abbreviations, the test statistic, and interpretations of each measure (i.e. score range and if higher/lower score is better/worse).
Response: We amended tables within the result subsection according to Reviewer 2 comments and we added a footnote at the bottom of each table.
Reviewer 2: You need to provide the questionnaire used as a supplement for verification. There needs to be much more analysis of the use of an unvalidated questionnaire.
Response: The questionnaire was added as a supplementary file. The questions used in this study were used before in another study by Abbas et al. (Abbas J, Hamoud K, Jubran R, Daher A. Has the COVID-19 outbreak altered the prevalence of low back pain among physiotherapy students? Journal of American College Health. 2021:1-6.).
Reviewer 2: Your discussion would benefit from using sub-headings to organise your thoughts. Please place in subheadings defining the topic of your argument and improve the linkage between arguments.
Response: Thank you for the comment, subheadings were added to the discussion sections to define each topic in the discussion.
Reviewer 2: You need to include a limitations section - for which this study has some major limitations. These include, but are not limited to, lack of control group, missing pre-exercise and sedentary levels, unvalidated questionnaire, etc.
Response: A limitation section was added to the end of the discussion section.
Reviewer 2: The conclusion needs to mentioned more about future directions and areas for exploration.
Response: The conclusion was rewritten to include more about future directions.

Round 2
Reviewer 1 Report
The authors present a revised version of their manuscript investigating the effects of distance learning on low back pain prevalence in students. The initial manuscript has been strongly improved. The reviewer thank the authors for the associated work done. However, there is still some minor points that need to be addressed to the authors for correction before publication.
MATERIALS AND METHODS
2.2 Questionnaire. Please insert that the questionnaire used and revised by experts is inspired from Abbas et al. (2021) and add the reference in the section “References”.
RESULTS
Please, refer all tables (1 to 7) to appropriated subsections. For example, Table 1 should be referred at the end of the first sentence of the subsection Demographic data: “A total of 84 students participated in the study, 46 (54.8%) females and 38 (45.2%) males (Table 1)”.
Table 6. Third column, line 2. There is something wrong about the reported % (200%). This should be 8.33%, 6.25%, 83,33% and 2.09%, respectively.
DISCUSSION
Please change Limitation for Limitations.
Author Response
Manuscript title: Low Back Pain Prevalence Among Distance Learning Students
Response to the reviewers’ comments:
Thank you again for your time and comments on this manuscript. We found these comments will improve the clarity and the readability of this research article. Please note that within the following paragraphs we presented a reply to each of the reviewers’ comments.
Reviewer 1:
Reviewer 1: The authors present a revised version of their manuscript investigating the effects of distance learning on low back pain prevalence in students. The initial manuscript has been strongly improved. The reviewer thank the authors for the associated work done. However, there is still some minor points that need to be addressed to the authors for correction before publication.
Response: We would like to thank again Reviewer 1 for his/her time and comments which helped us immensely in improving the overall quality of this manuscript. We again found his/her comments very constructive and precise; thus, the manuscript was amended according to Reviewer 1 comments and suggestions.
MATERIALS AND METHODS
Reviewer 1: 2.2 Questionnaire. Please insert that the questionnaire used and revised by experts is inspired from Abbas et al. (2021) and add the reference in the section “References”.
Response: We would like to thank Reviewer 1 for his/her suggestion. The text now read in page 2, line 88-90 as follows “An online-based questionnaire was developed by the investigators through reviewing literature with relevant objectives, the questionnaire used and revised by experts was inspired from Abbas et al. [21].”
RESULTS
Reviewer 1: Please, refer all tables (1 to 7) to appropriated subsections. For example, Table 1 should be referred at the end of the first sentence of the subsection Demographic data: “A total of 84 students participated in the study, 46 (54.8%) females and 38 (45.2%) males (Table 1)”.
Response: We thank Reviewer 1 for his/her comment on the Result section. We amended the Result section and referred all tables within this section to appropriate subsections.
Reviewer 1: Table 6. Third column, line 2. There is something wrong about the reported % (200%). This should be 8.33%, 6.25%, 83,33% and 2.09%, respectively.
Response: We are so sorry for these typo mistakes. We amended the third column according to Reviewer 1 comment.
DISCUSSION
Reviewer 1: Please change Limitation for Limitations.
Response: We are sorry for this typo mistake. We changed Limitation for Limitations.

Reviewer 2 Report
Thank you to the authors for systematically addressing my comments in their most recent revision.
Good luck with your future research into this field.
Author Response
Manuscript title: Low Back Pain Prevalence Among Distance Learning Students
Reviewer 2:
Reviewer 2: Thank you to the authors for systematically addressing my comments in their most recent revision.
Good luck with your future research into this field.
Response: We would like to thank again Reviewer 2 for his/her time and comments which helped us immensely in improving the overall quality of this manuscript. Thank you again.
